# The Full Value of Vaccine Assessments Concept—Current Opportunities and Recommendations

**DOI:** 10.3390/vaccines12040435

**Published:** 2024-04-18

**Authors:** Richard G. White, Nicolas A. Menzies, Allison Portnoy, Rebecca A. Clark, Cristiana M. Toscano, Charlotte Weller, Marta Tufet Bayona, Sheetal Prakash Silal, Ruth A. Karron, Jung-Seok Lee, Jean-Louis Excler, Jeremy A. Lauer, Birgitte Giersing, Philipp Lambach, Raymond Hutubessy, Mark Jit

**Affiliations:** 1Department of Infectious Disease Epidemiology, London School of Hygiene and Tropical Medicine, London WC1E 7HT, UK; rebecca.clark@lshtm.ac.uk (R.A.C.); mark.jit@lshtm.ac.uk (M.J.); 2Department of Global Health and Population, Harvard T.H. Chan School of Public Health, Boston, MA 02115, USA; nmenzies@hsph.harvard.edu; 3Center for Health Decision Science, Harvard T.H. Chan School of Public Health, Boston, MA 02115, USA; aportnoy@bu.edu; 4Department of Global Health, Boston University School of Public Health, Boston, MA 02118, USA; 5Department of Collective Health, Institute for Tropical Medicine and Public Health, Federal University of Goiás (UFG), Goiânia 74690-900, Brazil; ctoscano@terra.com.br; 6Wellcome, London NW1 2BE, UK; c.weller@wellcome.org; 7Gavi, 1218 Geneva, Switzerland; mtufet@gavi.org; 8Modelling and Simulation Hub, Africa, Department of Statistical Sciences, University of Cape Town, Cape Town 7701, South Africa; sheetal.silal@uct.ac.za; 9Centre for Global Health, Nuffield Department of Medicine, Oxford University, Oxford OX3 7BN, UK; 10Department of International Health, Johns Hopkins Bloomberg School of Public Health, Baltimore, MD 21205, USA; rkarron@jhu.edu; 11Policy and Economic Research Department, International Vaccine Institute, Seoul 08826, Republic of Korea; jungseok.lee@ivi.int; 12 International Vaccine Institute, Seoul 08826, Republic of Korea; jeanlouis.excler@ivi.int; 13Department of Management Science, Strathclyde Business School, Strathclyde University, Glasgow G1 1XQ, UK; jeremy.lauer@strath.ac.uk; 14Immunization, Vaccines and Biologicals Department, WHO, 1211 Geneva, Switzerland; giersingb@who.int (B.G.); lambachp@who.int (P.L.); hutubessyr@who.int (R.H.)

**Keywords:** vaccines, policy, World Health Organisation

## Abstract

For vaccine development and adoption decisions, the ‘Full Value of Vaccine Assessment’ (FVVA) framework has been proposed by the WHO to expand the range of evidence available to support the prioritization of candidate vaccines for investment and eventual uptake by low- and middle-income countries. Recent applications of the FVVA framework have already shown benefits. Building on the success of these applications, we see important new opportunities to maximize the future utility of FVVAs to country and global stakeholders and provide a proof-of-concept for analyses in other areas of disease control and prevention. These opportunities include the following: (1) FVVA producers should aim to create evidence that explicitly meets the needs of multiple key FVVA consumers, (2) the WHO and other key stakeholders should develop standardized methodologies for FVVAs, as well as guidance for how different stakeholders can explicitly reflect their values within the FVVA framework, and (3) the WHO should convene experts to further develop and prioritize the research agenda for outcomes and benefits relevant to the FVVA and elucidate methodological approaches and opportunities for standardization not only for less well-established benefits, but also for any relevant research gaps. We encourage FVVA stakeholders to engage with these opportunities.

## 1. Introduction

Major investments and policy decisions require a careful assessment of the consequences of each possible course of action, in comparison to available alternatives. For health interventions, this is often operationalized as an economic evaluation, in which the health outcomes of each alternative are compared to their costs, to identify the alternative that best balances the desire to improve health with the desire to best allocate limited resources considering the many competing potential uses.

However, traditional economic evaluation modalities such as cost-effectiveness analysis may not capture all the consequences of interest for a particular decision [1,2,3], and multiple alternative assessment frameworks—including budget impact analysis, extended cost-effectiveness analysis, distributional cost-effectiveness analysis, and multi-criteria decision analysis—have been developed as adjuncts to traditional economic evaluation. In particular, traditional cost-effectiveness analyses often fail to capture all of the outcomes of interest when it comes to immunization programs. Hence, various frameworks have been proposed that capture broader socioeconomic consequences of vaccines such as ecological externalities, macroeconomic effects, and insurance value [4,5,6,7,8]. The strengths and limitations of these approaches and frameworks for vaccine development and adoption decisions have been previously reviewed [2] in the context of a comprehensive ‘Full Value of Vaccine Assessment’ (FVVA) framework proposed by the WHO [9]. This framework is designed to expand the range of evidence available to support the prioritization of candidate vaccines for investment and eventual uptake by low- and middle-income countries, to not only report on traditional assessment criteria (for example, anticipated impact on disease-related deaths and cost-effectiveness) but include a wide range of health, economic, and distributional outcomes that may be relevant for a given vaccine candidate.

While FVVAs are often undertaken with the aim to inform investment in the development of vaccines, in particular their health economic outcomes can be used to inform country-level vaccine introduction and reimbursement decisions by national immunization technical advisory groups (NITAGs) and health technology assessment (HTA) agencies [10]. Such mechanisms for decision-making have been proposed since the 1970s as a means of synthesizing different streams of evidence about a health technology in order to inform its use [11]. Vaccine HTAs need to consider the value of a vaccine, so an FVVA is well-placed to give a holistic view of this aspect. However, HTAs also need to incorporate further elements that go beyond what FVVAs consider, such as ethical, legal, social, and political dimensions, as well as patient and public values. Another key instrument that WHO uses to assess the value of vaccines is the Vaccine Value Profile (VVP), which identifies research gaps that need to be addressed for an FVVA. VVPs have been published for a range of vaccines [12,13,14,15,16].

The rationale for the FVVA framework, as described by Hutubessy et al., Trotter et al., and others [2,9,10], is that these assessments will guide the evaluation and communication of the value of vaccines, facilitate alignment around priorities between stakeholders, and improve allocation of investments for vaccine development, policy-making, procurement, and introduction. The FVVA is designed to inform decisions at every point of the vaccine decision-making continuum, from discovery through early clinical development, large efficacy trials, licensure, broad scale implementation, and sustainability and from global to local decision-makers (see Figure 1 of [9]). The key principles of this approach include the adaption of existing assessment frameworks to include the additional types of benefit for which empirical evidence has become available in recent years, including non-health societal considerations such as equity and financial risk protection; a deliberative approach to evaluate vaccine candidates, which reflects the agency of each stakeholder; and the use of FVVA results to communicate the value of vaccines, through investment cases and business cases. While the motivations and principles for the FVVA approach will be relevant to many health topics, the problems that FVVAs attempt to solve are more prominent for vaccines. Externalities and market failures affect product development for many vaccine-preventable diseases, requiring coordinated action and cooperation to resolve.

Over the past years, FVVAs have become increasingly established, with the Group B Streptococcus (GBS) vaccine’s FVVA being the first example covering all elements of an FVVA in their entirety [17]. Since the publication of the GBS FVVA, substantial additional funding towards vaccine development has been granted. Beyond the publication, there is also indication that maternal immunization is now also a priority for BMGF, who are engaged in supporting the development of further evidence to explore the capacity in low-resource settings for the potential future introduction of RSV and GBS. At the time of writing, the FVVA framework has been applied to a range of different diseases and vaccines, including Group B Streptococcus, tuberculosis, Group A Streptococcus, measles–rubella micro-array patches (MAPs), and Shigella [17,18,19,20,21,22,23,24,25,26,27,28] (Table 1). Many of these vaccines continue to advance in development, with donors such as Gavi and the Vaccine Alliance, discussing potential mechanisms to fund their introduction. These assessments have collated existing studies as well as conducted new analyses to provide the evidence required for each vaccine. The FVVAs undertaken to date have had a beneficial impact by aligning and bringing together key stakeholders, accelerating vaccine development, engaging global stakeholders to discuss the value and feasibility of vaccines, providing information to country-level stakeholders, informing all participants about vaccine features that are of greatest value, and providing signals to funders about where investments will be most impactful. The identities of the stakeholders and how they contribute to the assessment have differed across FVVAs depending on what organizations and issues are most salient for a given vaccine candidate. However, for each FVVA, the analyses underlying the assessment have been undertaken by independent groups (typically academic institutions) to prevent the financial or other incentives faced by stakeholders from influencing analytic decisions.

Upcoming FVVAs are planned for non-typhoidal Salmonella and next generation influenza vaccines [30]. These FVVAs have reported evidence describing aggregate health benefits, the distribution of health benefits across population groups, financial risk protection, vaccine introduction costs, cost implications outside of vaccine delivery, implications for antimicrobial resistance and health security, budget impact, and macroeconomic consequences (Table 1). Each FVVA has covered different subsets of these domains, with variations in methods and definitions. These differences may be co-determined by what was technically feasible in a given disease, from prior expectations of where the major implications of a vaccine will incur, and from differing needs of the various stakeholders involved for each FVVA. While such differences are understandable, they also create challenges for comparing FVVA results. Moreover, while the paper by Hutubessy et al. [9] provides a conceptual framework for conducting FVVAs and the recent paper by Trotter et al. provides advice on its practical use [10], there is not yet concrete guidance on the optimal FVVA methods to be used, which would standardize the application of these assessments, facilitate fair comparisons, and allow larger themes or common evidence gaps to be identified. Building on the success of these prior applications and the FVVA framework [9,17,18,19,20,21,22,23,24,25,26,27,28], we propose three recommendations to continue the development and standardization of FVVA methods, in order to maximize the usefulness of future FVVAs for country and global consumers.

## 2. Current Opportunities and Recommendations

### 2.1. Addressing the Needs of Multiple Key FVVA Consumers

There will be multiple consumers of the evidence collated/created in an FVVA for a particular pathogen. This could include health ministries and ministries of finance staff, national and regional immunization technical advisory groups (NITAG, RITAG), national and sub-national implementors, multilateral investors (e.g., Gavi), global policy makers (e.g., the WHO), local and global regulatory authorities, manufacturers and R&D companies, civil society organizations, and funders (of research, vaccine development R&D, and/or vaccine introduction in countries; e.g., Wellcome, BMGF, USAID, US NIH). However, the evidence collated/created in a specific FVVA and the methods and key analytical assumptions used to create this evidence are unlikely to suit all these potential consumers because the types of evidence required, the methods used, and the analytic assumptions applied by FVVAs consumers may differ. Table 2 shows the evidence needs for decision-making reported by a single representative of a selection of key decision-makers: WHO Immunization and vaccines related implementation research advisory committee (IVIR-AC), WHO Product Development for Vaccines Advisory Committee (PDVAC), the funder and implementer GAVI, Regional Technical Advisory Groups on Immunization (RITAG) Pan American Health Organization (PAHO), the Brazilian and South African NITAGs, the Thai ministry of health’s Health Intervention and Technology Assessment Program (HITAP), and the product developer International Vaccine Institute (IVI) in Lao. As such, it cannot be considered comprehensive, but it does show some interesting commonalities and differences. Many evidence needs were shared by nearly all consumers, particularly those in the ‘health-related benefits to vaccinated individuals and society’, and the ‘community or health system externalities’ groupings. However, some evidence needs were much more variable between consumers, particularly those in the ‘broader economic indicators’ grouping. Therefore, the evidence generated for the FVVA is unlikely to match all the needs of the various key consumers because they will require different evidence types and/or different methods or assumptions to be used.

This could be addressed by requesting FVVA producers to generate evidence to explicitly meet the needs of multiple key consumers. For this, it would be neccessary for key consumers to be more explicit on the (a) the type of evidence they are likely to require and (b) any methods and analytic assumptions/parameters they would need applied.

The potential advantages of requesting FVVA suppliers to also create evidence explicitly matching multiple key consumer needs include that the evidence would be available earlier to support decision-making. It may also mean that fewer overall resources were needed from the global health community as many aspects of multiple analyses would likely be the same and would not need to be repeated if carried out in one study. FVVAs also have a comparative advantage for this purpose compared to Vaccine Value Propositions, which do not include specific requests for evidence generation. Ultimately, if appropriate evidence is available sooner, it could have important benefits such as avoiding the development of vaccines with likely low demand as it would be clear earlier that they would not meet decision-maker requirements. In other words, FVVA results can help enhance the efficiency of vaccine R&D, making vaccines more affordable and accessible to the target population. The potential disadvantages of requesting FVVA suppliers to also create evidence explicitly matching multiple key consumer needs include the potential misalignment of timing of decision-maker needs and therefore the extra evidence may not be useful when produced, e.g., the earliest FVVAs may be to primarily inform vaccine R&D and therefore the results would be less relevant to regulatory agencies or public funders unless the newly developed product has reached clinical testing s. Also, the evidence necessary to generate additional outputs may not be available locally and at the time of analysis, resulting in delays while evidence is being sourced or incomplete analyses. Further, although fewer resources may be required overall compared to (potentially) multiple separate studies, a single evidence generation effort to suit many stakeholders may require more resources than a single evidence generation study focusing on just one consumer, and these funds may not be available. 

A recent example of where improved alignment may have been desirable is the recent WHO TB FVVA [18,19,20,21,22] and Gavi Vaccine Investment Strategy (VIS) [35]. In 2020, the request for applications (RFA) for the WHO Full Value of TB Vaccine Assessment was awarded to a group of researchers, including authors on this commentary. The RFA was created with reference to the Gavi evidence needs, but despite these best efforts, when the Gavi vaccine investment strategy (VIS) needed evidence in 2023/4, a second evidence generation study was required, the authors of which are also included in this commentary. Because of the relatively short period of time between the two exercises, it was possible to use the models created for the WHO FVVA as a basis for the Gavi VIS, leading to some resource efficiencies. These savings could have been increased, with additional alignment between the WHO FVVA and the Gavi VIS processes, for example on (a) the need for the vaccine impact per fully vaccinated person outcome to be calculated [36], (b) the range of different countries to be included in the analyses, and (c) assumptions on different coverage values, vaccine characteristic values, and implementation strategies. Of these differences, we believe at least the first could have been known at the time of WHO FVVA analysis, and in addition, if the countries, vaccine coverages, and implementation strategy assumptions Gavi would likely require had been available at the time of the WHO FVVA, then it may have been possible to create and run these scenarios at the time of the WHO FVVA.

Meeting the needs of multiple stakeholders can not only be achieved by key consumers being explicit on the type of evidence/methods/assumptions they require, but it can also be achieved in other ways. For example, the influenza FVVA attempted to link R&D and country decision-making by assessing the vaccine development pipeline and the country preferences towards a next generation vaccine identified through MCDA analyses.

To address these opportunities, we recommend FVVA producers should aim to create evidence that explicitly meets the needs of multiple key FVVA consumers.

**Recommendation 1:** FVVA producers should aim to create evidence that explicitly meets the needs of multiple key FVVA consumers.

### 2.2. Addressing Standardized Methodologies and Guidance for How Different Stakeholders Can Explicitly Reflect Their Values within the FVVA Framework

The WHO and collaborators have developed a framework for FVVAs [9,10] that builds on several previous streams of the WHO-led work, including the FVVA’s predecessor Full Public Health Value Propositions for Vaccines (FPHVPs) and research around the broader economic impact on vaccines (BEIVIP). These streams of work have enabled the generation of a catalogue of outcomes that are of value to stakeholders, but are underexplored in most vaccine evaluations, such as the impact on antimicrobial resistance, equity, macroeconomic growth, household human capital development, and carbon emissions, e.g., [3]. The FVVA framework also delineates the principles of engagement that were designed to be aligned with the principles of the Immunization Agenda 2030 [37], such as being country-owned and partnership-based. The framework also describes, although does not prescribe, different assessment methods that can be consistent with these outcomes and principles, including (extended) cost-effectiveness analysis, benefit–cost analysis, multi-criteria decision analysis, and economic surplus analysis. Lastly, the framework also considers broader market factors such as consumer demand, service delivery, and uptake issues.

What remains is to operationalize this guidance in a way that enables analysts to produce routine evaluations consistent with the FVVA framework, without the need to conduct their own methodological research. The framework would also benefit from being adapted to the needs of various stakeholders that may want to use FVVAs for decision-making (including the WHO, global partners such as Gavi and BMGF, country NITAGs and governments, and development agencies). This process is outlined in Recommendation 3 and requires addressing two challenges.

First, well-developed methods and guidance exist for the traditional aspects of value, such as the direct health impact of vaccines and health-care service costs. For other more novel aspects, new methods have been explored and proposed in research publications but have yet to be standardized, so that they can be routinely used to generate evidence in FVVAs. For instance, multiple methods exist for incorporating equity concerns into economic evaluations [38], but NITAGs and HTA agencies have not formally endorsed any of these methods as a way to consider equity in health economic considerations. This standardization will be useful and will have advantages, for example, in minimizing methodological errors, but will also face challenges such as being operationalizable given differing data availability and health and economic circumstances in each application.

Second, FVVAs need to reflect the normative values of their users, but these values are usually not explicitly defined. For instance, stakeholders may place different weights on the trade-off between efficiency (greater health impact per dollar spent) and equity (prioritizing health and financial benefits to the worst off). An expensive vaccine campaign that reaches a remote and underserved population may be attractive to a stakeholder that prioritizes equity, while another stakeholder may prefer to support a vaccination program that can reach a greater number of people for the same cost. Explicitly defining the quantitative trade-offs inherent in their values would aid analysts in adapting results for different stakeholders, as well as enabling greater transparency and scrutiny around investment priorities.

**Recommendation 2:** WHO and other key stakeholders should develop standardized methodologies for FVVAs, as well as guidance for how different stakeholders can explicitly reflect their values within the FVVA framework.

### 2.3. The Next Research Agenda for the FVVA Framework

To address the above gaps and inform a future FVVA research agenda, we recommend that the WHO convenes an independent multi-disciplinary expert consultation to shed light on the important sources of evidence for decision-making across the vaccine development, introduction, and sustained implementation process, to develop standardized methodologies for FVVAs, andguidance for how different stakeholders can explicitly reflect their values within the FVVA framework (e.g., [39]). These needs can then be operationalized into analytic frameworks for evidence generation, specifically as a comprehensive list of potential outcomes of interest to relevant FVVA stakeholders, along with potential metrics for evaluation and opportunities for standardization. This expert convening would assess what outcomes and methods can be standardized and what outcomes and methods require additional investigation and understanding. Expert feedback should be sought as well on the interpretation and implications of standardizing analytic frameworks for future implementation of and comparison between FVVA efforts. In other words, the methodological approaches appropriate for standardization may depend on what are we hoping to achieve with FVVA application generally compared to what is specific to an individual disease area application.

The consultation could highlight areas where standard methodologies cannot be proposed because the techniques are not yet well-developed, as this could spur further research in these areas and appropriate sources of input evidence and proxies to use where these are not readily available. The consultation could also discuss how these methods can be integrated within a variety of assessment methods (e.g., cost-effectiveness analysis, benefit–cost analysis, multi-criteria decision analysis), particularly highlighting points of standardization that are relevant across frameworks. For instance, willingness-to-pay values are relevant not only to cost-effectiveness analysis but also for monetizing health gains in benefit–cost analysis and return on investment analysis, as well as identifying swing weights in multi-criteria decision analysis. As the methodologies develop, the framework can continuously be reviewed by appropriate WHO committees such as IVIR-AC and PDVAC.

Some cross-disciplinary areas that may need further research to develop include productivity gains, antibiotic usage and resistance, macroeconomic impact, climate change, and service delivery. Productivity gains can be attributable to multiple factors and therefore estimated with multiple metrics. In addition to productivity gains due to years lived with disability and years of life lost averted by vaccination (typically measured as a disability-adjusted life year in LMICs), productivity gains can also be due to improved cognition and educational attainment, which may be evaluated with longitudinal metrics on educational outcomes and lifetime earnings. The productivity gains resulting from improved health and survival of both children and adults can also contribute to economic improvements through changes in household choices such as fertility and consumption/savings. Potential metrics for these types of gains could include female labor force participation, household investment per child, and the dependency ratio. When considering health improvements in the community through reduced antibiotic usage, transmission dynamic models could be leveraged to estimate the prevalence of antibiotic resistance. The estimation of macroeconomic impact with general accounting or macroeconomic models could follow several methodological approaches, including changes to individual net transfers to the national budget over the lifetime or changes to national income or production because of long-term changes to the labor supply, as measured by metrics such as return on investment, net present value of investment, or change in GDP (or GDP per capita). Given important linkages between climate change and population health, it may be relevant for applications of FVVA to estimate outcomes such as reducing the environmental impact of health-care systems through averted hospitalizations and outpatient visits, improving individual-level health through strengthened immunity, and epidemic preparedness for circulation of emerging infectious diseases globally. For service delivery outcomes, it may be important to quantify the infrastructure and human resources available to deliver immunizations, which can be critical information for the implementation of new programs in LMICs with a known capacity constraint. Other examples of WHO vaccine technical reports include [40,41,42,43,44,45,46,47,48,49].

The outcomes and benefits relevant to FVVA need to be in line with the methodological standards for the evaluation of other public health interventions, where relevant. In addition, other FVVA research needs not widely reached by current well-established methods for evidence generation merit further understanding and development of analytic frameworks and outcomes, such as country-level stakeholders, manufacturers, R&D funders, and non-vaccine HTA experts. Upfront investment to better understand the needs of these stakeholders will be important to ensure that FVVA applications can be flexible and nimble in their responsiveness to stakeholder needs in each disease area, as well as accepted by wider research communities working on health technology assessments.

As such, we recommend the WHO convenes experts to further develop and prioritize the research agenda for outcomes and benefits relevant to the FVVA and elucidate methodological approaches and opportunities for standardization not only for less well-established benefits, but also for any relevant research gaps.

**Recommendation 3:** The WHO should convene experts to further develop and prioritize the research agenda for outcomes and benefits relevant to the FVVA and elucidate approaches and opportunities for standardization not only for less well-established benefits, but also for any relevant research gaps.

All three recommendations should include methods for continuous feedback and iterative improvement of the research agenda and methodologies, ensuring that they remain relevant and responsive to new findings and changing health environments.

## 3. Conclusions

The FVVA framework has been developed, and recent applications have already shown benefits in generating evidence on non-traditional aspects of vaccine value. Building on the success of these applications, we see important new opportunities to maximize the future utility of FVVAs to country and global stakeholders and provide a proof-of-concept for analyses in other areas of disease control and prevention. These opportunities include the following: (1) FVVA producers should aim to create evidence that explicitly meets the needs of multiple key FVVA consumers, (2) the WHO and other key stakeholders should develop standardized methodologies for FVVAs, as well as guidance for how different stakeholders can explicitly reflect their values within the FVVA framework, and (3) the WHO should convene experts to further develop and prioritize the research agenda for outcomes and benefits relevant to the FVVA and elucidate methodological approaches and opportunities for standardization not only for less well-established benefits, but also for any relevant research gaps. In addition, there is an ongoing pathogen prioritization effort by the WHO, which could be a basis for future FVVAs. We encourage FVVA stakeholders to engage with these opportunities.

## Figures and Tables

**Table 1 vaccines-12-00435-t001:** Summary of the FVVAs that have been carried out and types of vaccine value evidence they generated.

	Group B Streptococcus (2021, [17])	TB (2022, [18,19,20,21,22])	Group A Streptococcus (2022, [23,24,25,26])	MR-MAP iFVVA, 2023, [27])	Shigella (2024, [28])
**Health-related benefits to vaccinated individuals and society**					
Health gains	✔	✔	✔	✔	✔
Health-care cost savings	✔	✔	✔	✔	✔
**Efficiency and budget impact**					
Cost-effectiveness analysis/benefit cost analysis (BCA)	✔	✔	✔	✔	✔
Return on investment	✔	✔	✔		
Budget impact	✔	✔			
**Productivity-related benefits**					
Productivity gains related to care		✔	✔		
Productivity gains related to health effects		✔	✔		✔
Productivity gains related to non-utility capabilities *					
**Community or health systems externalities**					
Ecological effects **		✔			
-Antimicrobial resistance (AMR)		✔	✔		
-Health security/preparedness ***		✔			
Equity		✔			
Financial and programmatic synergies and sustainability					
Household security					
**Broader economic indicators**					
Changes to household behavior					
Public sector budget impact resulting from improved productivity ****					
Macroeconomic impact		✔			
**Other**					
Market size	✔	✔	✔		✔
R&D spending requirements			✔		

Notes: A tick indicates that this type of evidence was generated in the FVVA. Categories adapted from Jit et al. [3]. * Most cost-effectiveness evaluations focus on maximizing individual preference-based measures of health. Capabilities refer to the ability of individuals to function in particular ways and offer an alternative way to assess the value of health-altering intervention [29]. ** For e.g., herd effects, serotype replacement. *** For e.g., pandemic preparedness, climate change preparedness. **** reflects the increased tax take due to improved productivity (easily confused with budget impact of intervention costs).

**Table 2 vaccines-12-00435-t002:** Summary of FVVA ‘consumer’ needs.

	WHO IVIR-AC [31]	WHO PDVAC [32]	Gavi [33]	RITAG PAHO	NITAG Brazil	NITAG South Africa	MoH/ HITAP Thailand	Product developer, IVI Lao [34]
**Health-related benefits to vaccinated individuals and society**								
Health gains	✔	✔	✔	✔	✔	✔	✔	✔
Health-care cost savings	✔	✔	✔	✔	✔	✔	✔	✔
**Efficiency and budget impact**								
Cost-effectiveness analysis/benefit cost analysis (BCA)	✔		✔	✔	✔	✔	✔	✔
Return on investment	✔			✔	✔			
Budget impact	✔		✔	✔	✔	✔	✔	✔
**Productivity-related benefits**								
Productivity gains related to care	✔	✔	✔		✔		✔	✔
Productivity gains related to health effects	✔	✔	✔		✔			✔
Productivity gains related to non-utility capabilities *					✔			
**Community or health systems externalities**								
Ecological effects **	✔	✔	✔	✔	✔		✔	✔
- Antimicrobial resistance (AMR)	✔	✔	✔	✔	✔			✔
- Health security/preparedness ***	✔	✔	✔	✔	✔			✔
Equity	✔	✔	✔	✔	✔	✔	✔	✔
Financial and programmatic synergies and sustainability		✔	✔		✔			✔
Household security			✔		✔			
**Broader economic indicators**								
Changes to household behavior								
Public sector budget impact resulting from improved productivity ****					✔			✔
Macroeconomic impact	✔				✔		✔	✔
**Other**								
Market size			✔					✔
R&D spending requirements								

Notes: These are responses from representatives of each stakeholder below and may not constitute the official position of the organizations. A tick indicates that the stakeholder needs this type of evidence. Categories adapted from Jit et al. [3]. * Most cost-effectiveness evaluations focus on maximizing individual preference-based measures of health. Capabilities refer to the ability of individuals to function in particular ways and offer an alternative way to assess the value of health-altering intervention [29]. ** For e.g., herd effects, serotype replacement. *** For e.g., pandemic preparedness, climate change preparedness. **** reflects the increased tax take due to improved productivity (easily confused with budget impact of intervention costs).

## Data Availability

Data is contained within the article.

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
