# Peer review of "The Full Value of Vaccine Assessments Concept—Current Opportunities and Recommendations"

_vaccines, 2024, doi:10.3390/vaccines12040435_

Round 1
Reviewer 1 Report
Comments and Suggestions for Authors
Overall, the manuscript is comprehensive and provides a clear roadmap for improving the FVVA framework's relevance and efficacy.
The text provides a detailed overview of the necessity and complexity of economic assessments in health treatments, with a specific emphasis on vaccine assessment utilizing multiple frameworks, including the Full Value of Vaccine Assessment (FVVA). While the introduction clearly establishes the background and importance of such reviews, its presentation could be faulted in the following areas:
1. The introduction lists several assessment frameworks but does not critically evaluate their relative efficacy or limitations. A more detailed explanation of each framework's merits and drawbacks.
2. While the introduction mentions the application of the FVVA framework to numerous diseases and vaccinations, it does not provide particular examples or empirical evidence demonstrating the efficacy or consequences of these assessments. Including such facts may bolster the case for the FVVA's utility and impact.
3. The authors discuss stakeholder alignment and the participation of various global and local entities in the FVVA process. However, it could go into further detail about how these stakeholders are involved, the nature of their contributions, and how their competing interests or viewpoints are addressed in the FVVA framework.
4. While there is a brief mention of upcoming FVVAs and the need for methodological standardization, a more in-depth discussion of the challenges associated with implementing and standardizing the FVVA approach could provide a more realistic picture of the framework's potential and limitations. This may involve correcting methodological errors, data availability concerns, and the FVVA's adaptation to various health and economic circumstances.
5. The introduction may benefit from a more detailed explanation of how the FVVA influences decision-making processes at many levels, from global policy to local implementation.
6. The recommendation should include methods for continuous feedback and iterative improvement of the research agenda and methodologies, ensuring that they remain relevant and responsive to new findings and changing health environments.
Comments on the Quality of English Language
The written English in the provided text is of high quality, demonstrating clear structure, advanced vocabulary, and formal tone appropriate for a technical or academic document. However, minor editing is needed.
Author Response
| Overall, the manuscript is comprehensive and provides a clear roadmap for improving the FVVA framework's relevance and efficacy. The text provides a detailed overview of the necessity and complexity of economic assessments in health treatments, with a specific emphasis on vaccine assessment utilizing multiple frameworks, including the Full Value of Vaccine Assessment (FVVA). While the introduction clearly establishes the background and importance of such reviews, its presentation could be faulted in the following areas: |
Thank you for your comments |
| 1. The introduction lists several assessment frameworks but does not critically evaluate their relative efficacy or limitations. A more detailed explanation of each framework's merits and drawbacks. | Thank you for this comment. We have now expanded the introduction to include citations to the literature discussing each of these other approaches, as well as adding some additional text on the broader socioeconomic impact of vaccines, together with a reference. Given the focus of the manuscript we were not able to provide a critical evaluation of the established evaluation frameworks. While we agree with the reviewer such a comparison would be useful, to do justice to this would require a substantial expansion of the scope and word count of the article. However, we have pointed readers to the Hutubessy et al. article which does provide a table listing the strengths and limitations of most of these frameworks in the context of a full value of vaccines assessment. (page 4) |
| 2. While the introduction mentions the application of the FVVA framework to numerous diseases and vaccinations, it does not provide particular examples or empirical evidence demonstrating the efficacy or consequences of these assessments. Including such facts may bolster the case for the FVVA's utility and impact. | There is no comprehensive register kept by any of the organisations represented here about the effect that FVVAs have had on the field. However, many of the vaccines for which FVVAs were conducted have now advanced in terms of funding and development. In addition, maternal vaccination in general has become a priority for agencies like BMGF, possibly at least partly as a result of FVVAs for vaccines like GBS. We have now expanded the introduction to discuss some of these possible impacts of FVVAs. (page 5) |
| 3. The authors discuss stakeholder alignment and the participation of various global and local entities in the FVVA process. However, it could go into further detail about how these stakeholders are involved, the nature of their contributions, and how their competing interests or viewpoints are addressed in the FVVA framework. | Thank you for this comment, we have expanded the introduction section to discuss the ways that stakeholders participate in the FVVA: (page 5/6) |
| 4. While there is a brief mention of upcoming FVVAs and the need for methodological standardization, a more in-depth discussion of the challenges associated with implementing and standardizing the FVVA approach could provide a more realistic picture of the framework's potential and limitations. This may involve correcting methodological errors, data availability concerns, and the FVVA's adaptation to various health and economic circumstances. | Thankyou for this comment. Whilst not extensive, these points have been added to the manuscript on page 9 |
| 5. The introduction may benefit from a more detailed explanation of how the FVVA influences decision-making processes at many levels, from global policy to local implementation. | That information is actually shown in Figure 1 of the original FVVA paper (Hutubessy et al.), but we have now added a sentence to summarise it in the Introduction, while referring readers to Hutubessy et al. for further information. (Page 5) |
| 6. The recommendation should include methods for continuous feedback and iterative improvement of the research agenda and methodologies, ensuring that they remain relevant and responsive to new findings and changing health environments. | Thankyou for this comment. This text has been added to page 13. |
Reviewer 2 Report
Comments and Suggestions for Authors
This manuscript presents high-level overview of FVVA, “Full Value Vaccine Assessment” framework and recommendation on the implementation strategy of FVVA.
The manuscript seems to be prepared as commentary thus do not think I should value the manuscript for design adequacy, methodology and control on the research.
Recommendations are described below.
1) Brief outline of FVVA, what is the difference from traditional framework, should be described in introduction.
2) Recommendation part should be appear in main body text section. Body text section should be differentiated from introduction section. Maybe assessments section (from line 4, page 5) may include some research and original consideration contents. As the manuscript seems to be commentary, some headings explaining the contents of paragraphs should be given. Or, headings (now, recommendation 1 thorough 3) should be replaced to headings describing episodes, and observation from episodes should be presented as recommendation.
Author Response
| R2 | |
| This manuscript presents high-level overview of FVVA, “Full Value Vaccine Assessment” framework and recommendation on the implementation strategy of FVVA. The manuscript seems to be prepared as commentary thus do not think I should value the manuscript for design adequacy, methodology and control on the research. Recommendations are described below. |
Thankyou |
| 1) Brief outline of FVVA, what is the difference from traditional framework, should be described in introduction. | Thank you for this suggestion. In the introduction we have expanded the description of the FVVA and how it differs from the traditional approach: (page 4) |
| 2) Recommendation part should be appear in main body text section. Body text section should be differentiated from introduction section. Maybe assessments section (from line 4, page 5) may include some research and original consideration contents. As the manuscript seems to be commentary, some headings explaining the contents of paragraphs should be given. Or, headings (now, recommendation 1 thorough 3) should be replaced to headings describing episodes, and observation from episodes should be presented as recommendation | Thankyou for this comment. We have moved the recommendation to the end of each section in the body text, and add a new title for each of these sections (pages 5, 8, 9, 10, and 11) |